



# Momentum fluxes from airborne wind measurements in three cumulus cases over land

Ada Mariska Koning[1], Louise Nuijens[1], and Christian Mallaun[2]

[1]Delft University of Technology
[2]Deutsches Zentrum für Luft- und Raumfahrt

**Correspondence:** Mariska Koning (A.M.Koning@tudelft.nl)

**Abstract.** This study combines airborne Doppler Wind Lidar (DWL) observations with high-frequency in situ wind measurements from a gust probe, a combination that to our knowledge has not been used before. The two measurement techniques show a similar mean in the wind components throughout the flights and are then used to study momentum transport in relation to shallow cumulus over land. We present three case studies ranging from forced cumulus humilis to thicker clouds associated

with stronger popcorn-like convection after a cold front passage. The wind profiles obtained with the DWL are helpful in explaining the momentum fluxes that are calculated from the 100 Hz in situ data using the eddy covariance method. Most of the momentum flux profiles revealed down-gradient momentum transport that was generally strongest within the mixed-layer and decreasing towards cloud tops. Comparing clear-sky and cloud-topped transects, the cloudy skies revealed a substantial enhancement in the mixed-layer momentum flux (more than twice as much). On one track during the third flight, after a post-

cold-front passage and displaying thicker clouds, shows a momentum flux profile that did not decrease linearly with height as expected from shear-driven small-scale turbulence. The momentum in the mixed layer was very small, but a very strong flux has been observed in the cloud layer. Moreover, the updraft contribution to the flux was much larger in this case than in all other tracks that have been flown during the campaign. Last, we look into how much flux the different scales contribute to the overall transport. There we find that the largest scales (up to 7 km) usually carry most flux. However, sometimes the larger

scales have opposite contribution to the flux than the scales smaller than 7 km, which can then result in a smaller or almost no net flux.

## 1   Introduction

Observations of the vertical profile of wind are valuable for reducing forecast errors and for advancing the understanding of processes that influence wind variability, including large-scale dynamics and small-scale processes. In this paper we com-

bine state-of-the-art airborne wind lidars combined with traditional in situ turbulence measurements to measure the profile of wind and turbulent wind fluctuations within cloud-topped boundary layers, in which thermally-driven (convective) plumes are thought to play an important role in transporting wind. By measuring wind profiles at levels beyond meteorological towers and ground-based operational Doppler wind lidars, we aim to investigate the role of convection and clouds in setting the profile of momentum flux.



On local and regional scales, the growing wind energy industry has boosted wind profiling observations in the lowest layers of the atmosphere through the deployment of Doppler wind lidars (DWLs). DWLs conventionally measure high-resolution wind to the top of the wind turbine or to hub height (the centre of the wind turbine's rotor, up to 250 m). Such measurements are used to understand turbulent wind fluctuations in the surface layer that are influenced by weather, terrain, turbine wake effects or shear across the rotor-swept area (Bakhshi and Sandborn, 2020; Banta et al., 2013; Iungo and Porté-Agel, 2013;
Krishnamurthy et al., 2013; Mann et al., 2010).

One source of wind in the surface layer is the (downward) mixing of momentum from higher levels. Convection and clouds play an important role in this process, because they extend the depth of the boundary layer, allowing it to tap in regions aloft with faster moving winds. This transport of momentum (momentum fluxes) by convective eddies (thermals) and through clouds is broadly called convective momentum transport (CMT). Like small-scale turbulence, CMT is an unresolved process in forecast
models that contributes to uncertainties in local wind predictions. However, unlike the turbulent wind fluctuations measured in the surface layer through most commercial DWLs, few high-resolution wind profiles extend beyond the surface layer (> 200 m) to target wind fluctuations and momentum transport through CMT. Nevertheless, the upper part of the boundary layer is crucial for energy transport in the vertical direction. Because observations of winds and momentum transport are scarce in this part of the boundary layer, we target to measure specifically this area.

Our understanding of turbulent wind fluctuations throughout the boundary layer largely stem from few in situ turbulence measurements during research aircraft fights at selected height levels. A seminal example specifically analysing momentum fluxes is the study by LeMone and Pennell (1976), who flew below and through cumulus fields near Puerto Rico. Their derived wind and flux profiles reveal that the momentum flux profile can take a very different shape depending on clouds overhead. In particular, they found that in field of organised cumulus clouds, the momentum flux profile does not decrease linearly with
height as one would expect if only small-scale shear-driven turbulence would play a role. This implies that the net effect of turbulent fluctuations can be to accelerate winds, rather than just decelerate winds through turbulent diffusion.

Large-eddy simulations (LES) also reveal very different momentum flux profiles even for similar convective cloud situations (Zhu, 2015; Schlemmer et al., 2017; Saggiorato et al., 2020). LES output suggests that turbulent fluctuations on scales larger than 400 m explain a considerable part of the momentum flux, in particular above the surface layer towards the mixed-layer
top and within the cloud layer. Recently, Dixit et al. (2021) suggest that the absence of mesoscale circulations in idealised periodic-boundary LESs lead to an underestimation of momentum flux.

In the CloudBrake campaign, we were interested in measuring the turbulent to mesoscale contributions to the wind and momentum flux profile. The campaign involved dual-aeroplane flights over Germany using the Falcon and Cessna research aircrafts from the German Aeospace Center (Deutsches Luft- und Raumfahrt e.V., DLR) in Oberpfaffenhofen. The Falcon flew
at high altitudes deploying a downward looking 2 micron Doppler wind lidar (DWL) as well as the sideward looking ADM Aeolus demonstrator (A2D) lidar. When interested in the performance comparison between the satellite Aeolus measurements and those form the DWL, we suggest reading Witschas et al. (2020) and Lux et al. (2020). The smaller Cessna aeroplane took in situ wind and turbulence measurements at legs within the boundary layer. The three collocated flights captured conditions ranging from fair-weather shallow cumulus developing over hilly terrains to pre- and post-frontal convection with larger





cloudiness. Robust statistics cannot be applied in most airborne measurements, due to the small sample, changing conditions, and very different cases. However, by interpreting these cases we expect to gain valuable insights into the wind mixing and momentum transport under different conditions, as many previous flight campaigns have also proven (LeMone and Pennell, 1976; MacPherson and Isaac, 1977; Nicholls and LeMone, 1980; Rauber et al., 2007; Večenaj et al., 2012, e.g.)

In this paper we will describe the different meteorological conditions, the collected data and a comparison of the measured
winds and turbulence as measured remotely and in situ. The specific questions we aim to answer are:

1. How well do the DWL wind profiles match with the in situ observations?

2. How do the measured momentum flux profiles look like in the sub-cloud layer and in the cloud layer and are they in line with our expectation from theory?

3. Which scales contribute to the momentum flux?

We start with a description of the flight strategy, measurement techniques in section 2. This section also includes a short explanation of the expectations from theory and of the updraft detection method. Section 3 shows the meteorological conditions during the three flights. Section 4 explores the different momentum fluxes in relation to the wind profiles and cloudiness conditions. The contributions of updrafts are shown as well as the scales at which most momentum is carried. Thereafter the conclusions are presented.

## 75  2   Flight measurements and data processing

### 2.1   Flight strategy and measurements

The CloudBrake measurement campaign took place in Germany at the end of May and beginning of June, a period that is known to often display (shallow) cumulus clouds. Starting from around noon until 13:30 or 14:30 local time (CEST), the flights targeted a time of the day during which cumulus clouds are typically well developed. An impression of the cumulus and
weather conditions during each flights is given in Figure 1. The first flight on May 24 (2019) was a typical shallow cumulus day: starting out with clear skies and weak winds, local shallow cumulus started forming over the hilly parts of the Swabian Jura. Clouds remained shallow, reaching a thickness of approximately 500 m. The second flight was under the influence of an approaching cold front, providing an interesting and dynamic mixture of shallow cumulus- and stratocumulus-topped boundary layers. Above the shallow cumulus that were around 1 km thick, mid-level alto-cumulus and stratus layers were present. The
third flight, on June 4th 2019, experienced post cold front conditions. There was a large cumulus field with very diverse cloud tops. Clouds were at most 800 m thick and were typically thicker at the northern part of the leg.

During the 2–2.5-hour-flights, the two aeroplanes flew back and forth across pre-defined tracks. Flight legs ranged from 50 to 100 km in length to ensure sufficient low-frequency wind variability. During some of the flights the tracks were adjusted to ensure cumulus clouds were captured. Turbulence measurements using an in situ (3D) turbulence probe aboard the DLR
Cessna Grand Caravan were taken along that track at four different altitudes: within the mixed-layer, near cloud base, within





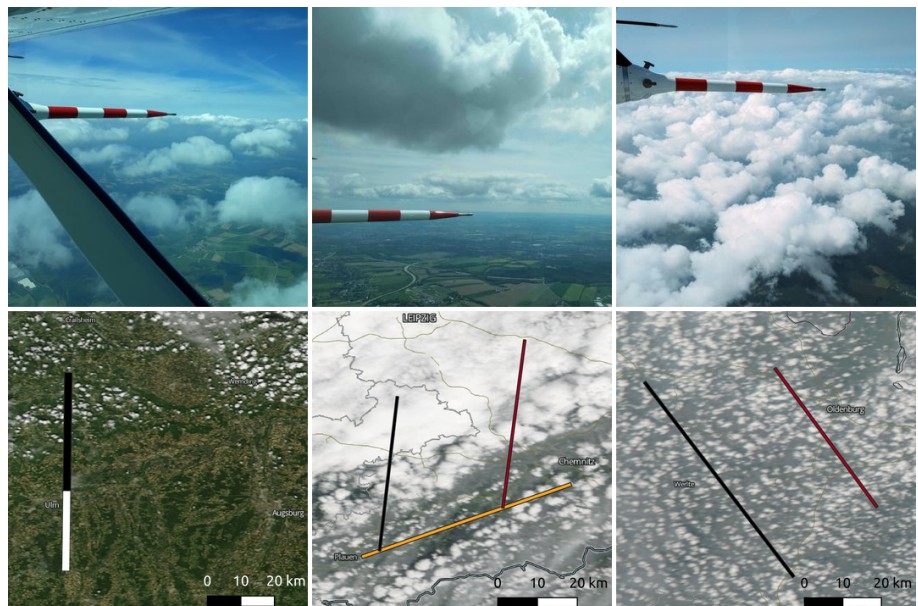

**Figure 1.** Photographs (upper row) and Modis satellite images from NASA Worldview Snapshots (lower row) of the cumulus fields during the flights on 2019-05-24 (left), where white is the cloud free area and black the cloudy area, 2019-05-27 (middle), where the west track is indicated in black, the east track in red and the southern track in yellow, and 2019-06-04 (right), where the west track is indicated in black and east in red. In each satellite picture, the horizontal black and white bar indicates a total distance of 20 km.

the cloud layer and near cloud top. Employing the downward staring Doppler wind LiDARs, the DLR Falcon remained around 11 km altitude throughout the flight. The instruments are described next.

### 2.1.1 In situ turbulence probe

The DLR Cessna Grand Caravan was equipped with (i) a meteorological sensor package (METPOD) that measures tem-
perature, humidity, pressure, and wind, and (ii) the IGI systems' AEROcontrol system, which combines measurements of a Differential Global Positioning System (DGPS) with a high-accuracy inertial reference system (IRS). Calibration of the devices before the flight and applying corrections afterwards result in a horizontal wind measurement uncertainty of 0.3 m s$^{-1}$ and 0.2 m s$^{-1}$ for the vertical wind component. Further details on the instrument specifics, calibration, correction procedure, and uncertainties can be found in Mallaun et al. (2015).
The high-frequency 100 Hz wind measurements, taken with a boom-mounted Rosemount model 858 AJ air velocity probe, are used for flux calculations. The aircraft movements are corrected using IGI. A linear fit is subtracted from the data before flux calculations. All scales from $10^{-2}$ Hz are included in this calculation, unless stated otherwise.





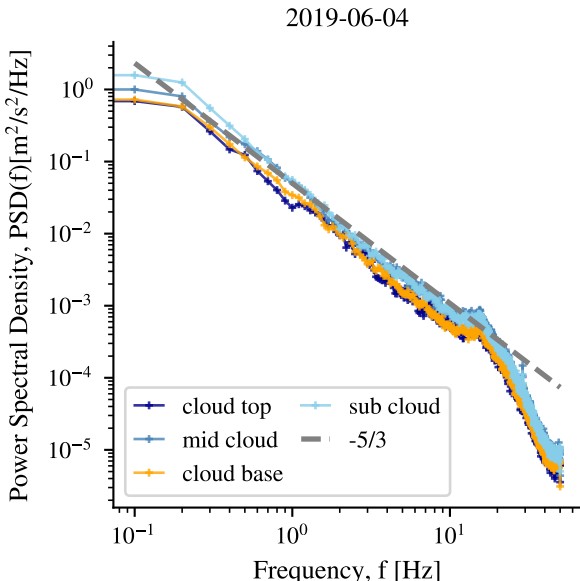

**Figure 2.** Power spectral density of the u-wind component for the western legs flown on 2019-06-04. Each altitude is represented by a different colour. The dashed line represents the -5/3 slope corresponding to the inertial sub-range.

### 2.1.2 Energy spectra

To check the quality of the measurements, we calculated the power spectral density (based on the Fast Fourier Transform), after
subtracting a linear trend from the data. Welch method was used with a Hann window with 10000 samples and 50% overlap to reduce noise in the spectrum. The spectrum of the $u$-wind component for one of the flight days is shown in Figure 2. Each line denotes a flight leg, whereby the legs flown in the sub-cloud layer and in cloud layer (light blue and medium blue, the latter mostly hidden behind the former) generally contain more energy than the legs flown near cloud base and cloud top (yellow and dark blue). Comparing the three wind components (not shown), turbulence appears to behave anisotropic: from 0.01-1 Hz,
$w$ contains more energy than $u$ and $v$. Between 1-10 Hz, $u$ has most energy, and $w$ least. The characteristic 5/3 slope of the inertial sub-range (dashed line) is seen from $\sim 0.2$ - 15 Hz (equivalent to a spatial resolution of 350 m down to 5 m, assuming a typical cruising speed of 65-75 m s$^{-1}$). From 15 Hz onward, the dampening of the fluctuations in the tube becomes visible and the signal falls of faster, except for one peak at 30 Hz, which is attributed to propeller effects (Mallaun et al., 2015).



### 2.1.3 Eddy-covariance fluxes

The time series are partitioned in leg-averaged values $\overline{\phi}$ and fluctuating parts $\phi'$ conform the Reynolds averaging technique. Fluxes and variances are then calculated by multiplying and averaging the fluctuations of $w$ and $\phi$ over a specific time window, known as the eddy-covariance method. For instance, the leg average flux of $\phi$ is given by:

$$\overline{w'\phi'} = 1/N \sum_{i=1}^{N} w_i' \phi_i' \qquad (1)$$

The smallest resolved frequency depends on the length of the leg, *i.e.* on the number of samples $N$: $f_{min} = f_s/N$, in which 120 $f_s$ is the sampling rate in Hz. Flying at a cruising speed of $\sim 65\text{-}75 \text{ m s}^{-1}$ at a constant height, and with constant ground speed, it is reasonable to assume that a static turbulent field is sampled. However, the statistical representation of the low frequencies is poor and therefore needs cautious interpretation.

### 2.1.4 Airborne Doppler wind LiDAR

Doppler wind LiDARs (DWLs) are the international standard for wind measurements and have been used for among other 125 things 1) data assimilation experiments (Horányi et al., 2015; Pu et al., 2017; George et al., 2021, e.g.,), 2) to study for instance turbulence, gravity waves, orographic effects (Yuan et al., 2020; Gisinger et al., 2020; Baidar et al., 2020, e.g.,), and 3) to monitor the flow in wind farms (Käsler et al., 2010; Wagner et al., 2017; Zhan et al., 2020; Schneemann et al., 2021, e.g.,). The coherent detection DWL employed in this study has a wavelength of 2022.54 nm (approximately 2 $\mu$m), being eye-safe and operating in the Rayleigh scattering regime. The (vertical) resolution of the wind measurements depends on both the duration 130 of the pulse, also called pulse width, and the distance that the signal can travel during the sampling time. The shorter the pulse, the better the spatial resolution, although a reasonable sampling duration is needed to ensure sufficient accuracy of the velocity estimation (Liu et al., 2019). With a pulse width of $\sim 400$ ns and an averaging time of 1 s, we have a vertical resolution of 100 m (Witschas et al., 2017). Furthermore, the aircraft speed influences the horizontal resolution. Flying with approximately 200 m s$^{-1}$ and having a sampling frequency of $\sim 40$ s, the horizontal resolution is about 8 km. Pulsed LiDARs have a blind spot 135 of tens to hundreds of meters near the beam source, depending on the pulse duration and range gate width (Liu et al., 2019). Therefore, although flying at 11 km, the first wind velocities are obtained from approximately 7 km altitude down to about 500 m. The DWL employed in this study has previously been compared to dropsonde measurements, in which the systematic error has been found to remain below 0.1 m s$^{-1}$ and the random error to vary between 0.92 and 1.5 m s$^{-1}$ (Weissmann et al., 2005; Chouza et al., 2016; Schaefler et al., 2018; Witschas et al., 2020).

140 The Velocity-Azimuth Display technique (Browning and Wexler, 1968) with an off-nadir angle of 20 degrees, is used to retrieve all three wind components. The processing algorithm that is applied to retrieve the wind vectors from one revolution of line-of-sight measurements is described in Witschas et al. (2017).

Figure 3 shows an example of the wind anomalies (i.e. the wind measurements of which the average wind during the measurement flight is subtracted) on June 4 2019. The turning points between legs are indicated with white vertical lines,



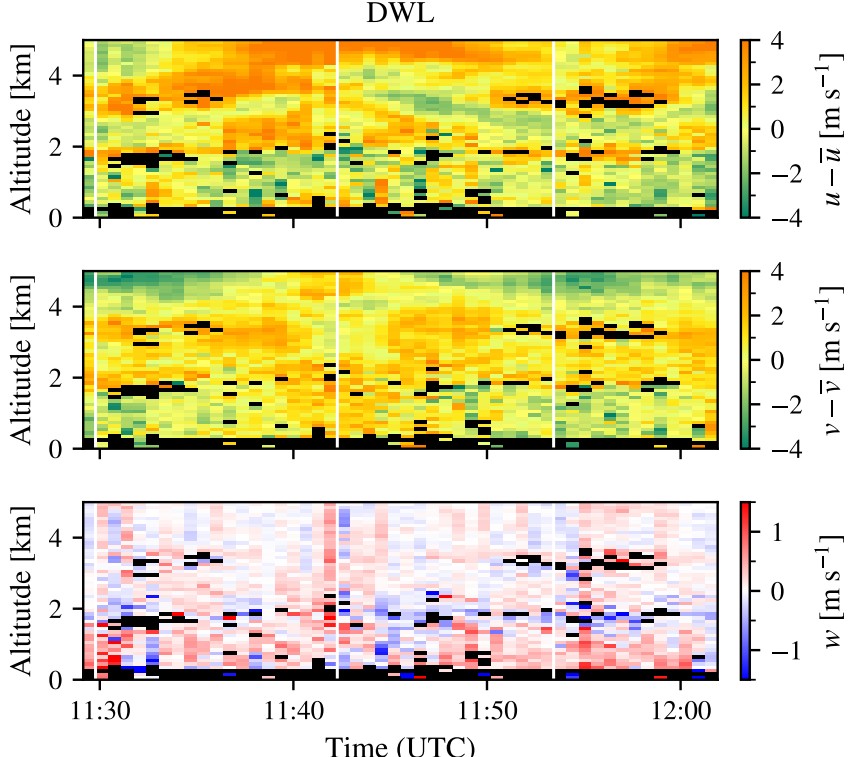

**Figure 3.** Anomalies of zonal (u), meridional (v), and vertical (w) wind measurements from the DWL, zoomed in on the lowest 5 km. Measurements taken on 2019-06-04. Missing values are indicated in black and often correspond to clouds (1-2 km altitude). White vertical lines indicate turning points on the track.

revealing similar but mirrored wind structures on subsequent legs. On this particular flight, the track was moved further to the east around 11:40 UTC, where different structures are visible. Data gaps, which can be associated with clouds, are indicated in black.

The top of the boundary layer that is around 2 km altitude is clearly visible in the $w$ fluctuations, with larger fluctuations below, and smaller above. The top of the boundary layer is marked by predominantly blue colours, indicating negative velocities produced by overshooting thermals that become negatively buoyant. Within the boundary layer updrafts generate the largest fluctuations, while a few downdrafts extending to the surface are also evident. It appears that the DWL can at least to some extent observe the coherent convective features that are responsible for mass transport of scalars and momentum.

For one of the legs in Figure 3, the histograms of the sub-cloud layer $u$, $v$, and $w$ wind are compared in Figure 4. Mean horizontal winds over this leg are comparable, although slightly overestimated, but despite the much coarser resolution and missing $v$ winds $< 2.5$ m s$^{-1}$, the wind variance observed by the DWL is only slightly overestimated. This gives us confidence that the DWL can provide complementary information of the (horizontal) wind profile at heights where in situ measurements

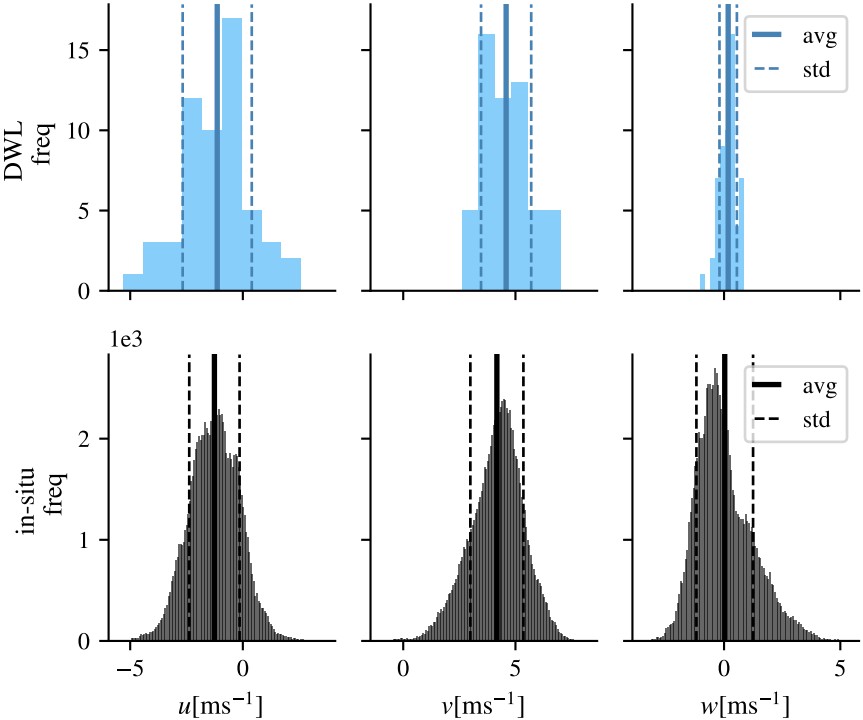

**Figure 4.** Distribution of $u$, $v$, and $w$ wind in the sub-cloud layer of the western track at 617 m altitude on 2019-06-04, as measured by the Doppler Wind LiDAR (top panels, blue) and the in situ turbulence probe (lower panels, black). The DWL range bin closest to the in situ flight height have been used.

are absent. It also tells us that horizontal wind fluctuations are dominated by scales larger than 1-2 km (the effective horizontal resolution of the DWL is $\sim 8.4$ km). On the other hand, the vertical wind shows much less variation than the in situ measurements. This is explained by the much larger area that is measured by the DWL: it can only see the average vertical velocity in this area, which on average is much lower than the vertical velocity of vertical transient small eddies than can be better captured by the in situ measurements.

## 2.2 Updraft detection algorithm

Using conditional sampling we identify updrafts, following the method described and tested by Lenschow and Stephens (1980). We conditionally sample on updrafts ($w' > 0$ & $w > 0$) that are wider than 100 m, and that have an excess in absolute humidity $\rho'_v > 0.5 \, \sigma_{\rho'_v}$. This method is more robust than using virtual temperature or buoyancy, and can be applied both in the sub-cloud and cloud layer.

Table 1 shows the updraft statistics of the legs flown on 4 June 2019. It lists the number of updrafts, the relative length of the leg that they occupy, the average horizontal size and the average updraft velocity. We find that the fraction of the leg



**Table 1.** This table shows the number of updrafts, relative updraft area, average updraft size, and average updraft speed for the legs flown on 4 June 2019.

| Updraft statistics | Number of updrafts | Updraft area [% of leg] | Chord length [m] | Updraft velocity [m s$^{-1}$] |
|---|---|---|---|---|
| **West (thicker clouds)** | | | | |
| Cloud top | 8 | 2.3 | 264 | 1.6 |
| Cloud layer | 12 | 5.4 | 412 | 2.2 |
| Cloud base | 20 | 8.1 | 328 | 1.4 |
| Mixed layer | 16 | 8.2 | 333 | 1.7 |
| **East (thinner clouds)** | | | | |
| Cloud top | 1 | 0.6 | 372 | 2.1 |
| Cloud base | 3 | 1.8 | 412 | 1.5 |
| Mixed layer | 10 | 4.7 | 289 | 1.4 |

that is covered by updrafts (updraft area) decreases with height, although the average updraft chord length (the length of the

updraft slice that we passed through) peaks at cloud base for the thinner clouds on the eastern track on June 4th as well for the clouds on 2019-05-24 (not shown). On May 24th, we find more and stronger updrafts in the cloud-topped mixed layer than under clear-skies, whereas updraft chord length is comparable. The largest average updraft velocity is found at cloud base, suggesting that the stronger mixed layer updrafts reach the lifted condensation level and benefit from the energy released at condensation. On June 4th, fastest average updraft speeds are found in the cloud layer in the case of thicker clouds. With the

thinner clouds fastest updraft speed is found at cloud top, although we must be careful as this includes only one sample.

## 3  Flight conditions: wind and thermodynamic profiles

Although the three flight days all captured a shallow cloud regime, they differed substantially in their characteristics of the wind (wind speed, wind shear and directional shear), providing a set of diverse case studies, whose wind and thermodynamic profiles are described next. For the wind, an entire profile of the mean and variance are shown from the DWL, with the in situ

measurements denoted on top (Figure 5). Except for the wind direction on May 24th, which varied greatly in the sub-cloud layer, the mean DWL and in situ winds compare very well.

The first flight (May 24) took place after a number of overcast days and heavy rain. Southern Germany was under influence of a broad area of high pressure west of Europe and over the Northern Atlantic and the conditions were very stable with hardly any clouds in Southern Germany. The northern part of the leg was flown over the Swabian Alps, where numerous

gliders were making use of the thermal structures that typically develop here and shallow cumulus with cloud bases near 2 km (dashed horizontal lines) and tops near 2.5 km developed. These were the focus of our measurements. Winds were weak and reasonably well mixed up to 1400 m, topped by a layer with strong wind turning near 1.5 km (some 500 m below cloud



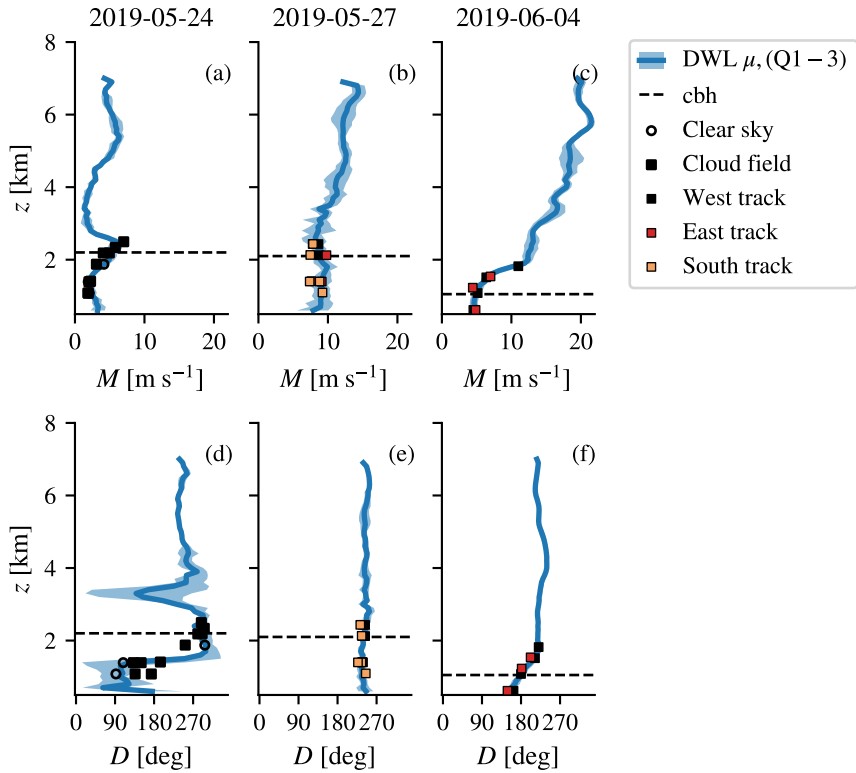

**Figure 5.** Average wind speed and wind direction profiles for each flight date. Average DWL profile indicated in blue, shading indicates the range between the first and third quartile. Average in situ measurement indicated with squares (measurements in/below cloud field areas) and circles (cloud free areas). Cloud base height (cbh) has been estimated during during the flight and is indicated with a horizontal dashed line.

base), and wind speed increased up to 2500 m in a layer extending through cloud base (Figure 5). In contrast, temperature and humidity were very well-mixed vertically. The atmosphere was relatively dry, with a pronounced inversion in temperature and
moisture starting near 2.2 km (Figure 6).

Considerably stronger wind speeds, but far less wind shear were present during the second flight (May 27) when we sampled air masses ahead of a cold front located SW-NE across eastern Germany (Figures 1(b), 5 (b,e)). The air masses were somewhat warmer and moister, but with a thermodynamic structure and a cloud base very similar to that of the first flight (Figure 6. Besides shallow convection, there was plenty of mid- and upper levels cloud, which we encountered at the end of the first flight
leg towards the north. Later, the front seemed to break up and skies were clearer, especially towards the southeast. Eventually, also in the southeastern area of our operations, shallow cumulus made way for stratocumulus layers, with only rare sights of clear sky and sunshine.

During the third flight (June 4th), we measured an extended field of shallow cumulus clouds that developed behind a cold front over northwestern Germany in air masses that were considerably colder and moister (Figure 6 (c,f)), with much lower



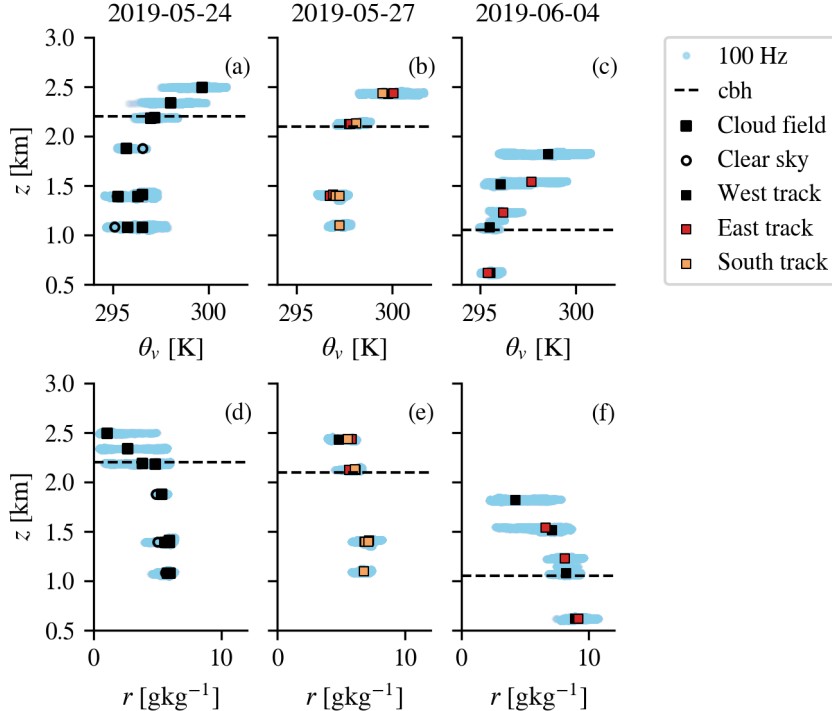

**Figure 6.** Virtual temperature and mixing ratio during the three flightdays. On the first flight there were three legs that were partly below clear-sky and partly below cloudy sky. They have been separated and are represented by open circles and closed squares, respectively. The other two cases had multiple tracks that are indicated with different colours. The raw 100 Hz data is indicated in light blue. Cloud base is indicated by the dashed horizontal line.

cloud bases near 1000 (western N-S leg) and 1200 m (eastern N-S leg) and very diverse cloud top heights (see Figure 1(c)), with maximum tops near 2 km. The cloud field was organised in patches of alternating cloudy and cloud-free air masses. As the clouds were getting deeper towards the northern parts of the leg, the relative sizes of the patches increased. Near- surface winds were weak and from the south, with strong shear and a turning from southeasterly to southwesterly winds right around cloud base (Figure 5(c,f)).

Based on the wind profiles, the three flights could be classified as having weak wind and strong shear either in the sub-cloud layer (Flight 1) or in the cloud layer (Flight 3), and having strong wind but little shear (Flight 2). In the next section, we will explore the associated turbulent statistics of these flights and evaluate whether the derived momentum flux profiles are in line with our expectations *e.g.,* that momentum fluxes throughout the mixed layer and cloud layer increase with wind shear as predicted by K-theory.





## 4  Momentum flux profiles

### 4.1  Sub-cloud and cloud layer profiles

In Figures 7 and 8 we consider the profiles of wind and momentum flux for the vector wind components $u$ and $v$ separately. As in Figure 5, the wind speed is shown for both the DWL (in blue) and the in situ turbulence probe at the flight levels (circles, squares). A guideline for the flux profiles in cloudy conditions are indicated with solid black lines, which are linearly interpolated between leg averaged values at the different flight levels (and are sometimes averaged over two legs the same level).

If shear-driven turbulent stress dominate the momentum flux, we expect the flux to behave as in K-theory or eddy diffusivity theory, which is mathematically expressed as:

$$\overline{u'w'} = -K\left(\frac{\partial \overline{u}}{\partial z}\right). \tag{2}$$

and similarly for $v$, in which $K$, the diffusivity coefficient, is strictly positive. K-theory is often used as a closure technique in models to denote so-called down-gradient momentum transport by small-scale turbulence, which refers to momentum being transport from regions with high to low momentum "down" the gradient, thereby acting to reduce the gradient.

On May 24th, ignoring the strong gradients in $u$ below $\sim 700$ m, $\partial_z u > 0$. This implies that air parcels that are displaced upward ($w' > 0$) generally have a negative $u$ perturbation compared to their environment ($u' < 0$). According to Equation 2 this leads to $\overline{u'w'} < 0$. This holds generally for all flight days. Negative $u$ perturbations are in particular evident from the actual wind in air masses sampled within updrafts, which tend to be several m s$^{-1}$ slower (pink triangles in Figure 7 and 8). Similarly, the meridional momentum fluxes are also down-gradient. For example, the gradient $\partial_z v < 0$ above 1 km on May 24th, corresponding to a positive meridional momentum flux ($\overline{v'w'} > 0$), and $\partial_z v > 0$ on June 4th, corresponding to $\overline{v'w'} < 0$.

The profiles of $\overline{u'w'}, \overline{v'w'}$ reveal that larger fluxes are measured on May 24$^{th}$ than on May 27$^{th}$, in line with the stronger shear present in $u$ and (to a lesser extent) in $v$. Fluxes typically decrease towards the boundary layer height (cloud top or mixed-layer top in case of clear sky), as the variance and skewness of vertical velocity decreases towards the top of the boundary layer (Figure 9).

Evidently, on May 24$^{th}$ the momentum fluxes throughout the mixed-layer increased considerably from the first transect of the flight, which captured a dry convective boundary layer (open circles), to the second transect, which is when cumulus clouds developed on top of the mixed layer (filled squares), although both transects have comparable $u, v$ profiles (Figure 7, 8). The larger fluxes reflect the presence of stronger turbulent eddies. Especially just below and at cloud base, much larger variances are present in $u$ and $v$ and throughout the mixed layer in $w$ (Figure 9).

While during May 24$^{th}$, 27$^{th}$ and the eastern leg on June 4$^{th}$ the fluxes decreased towards cloud base, with little flux remaining in the cloud layer. The western leg on June 4$^{th}$ shows an increase in momentum fluxes with height (in particular $\overline{u'w'}$ but also $\overline{v'w'}$). Whereas the flux in the mixed-layer below clouds is almost negligible, one of the largest fluxes was measured in the cloud layer ($\overline{u'w'} \sim 0.4$ m$^2$s$^{-2}$). Clouds on this western leg had a lower base (just above 1 km) and higher cloud tops (up to





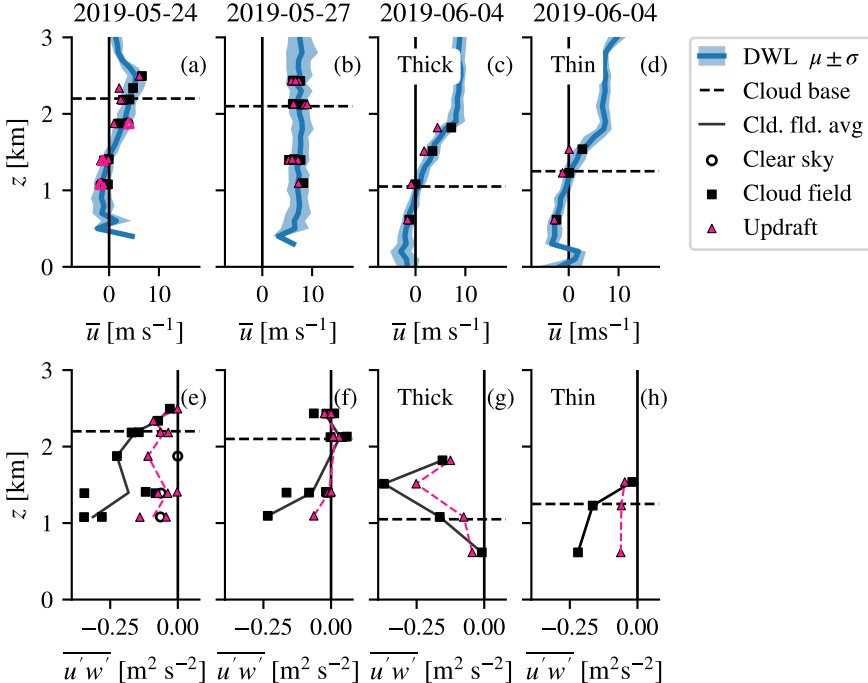

**Figure 7.** Average $u$ (a-d) and $\overline{u'w'}$ (e-h) profiles for each flight date. The u-axis is positive eastward. To obtain these profiles, we applied the same averaging procedure as in Figure 5 and the eddy covariance method as described in Section 2.1.3. On 2019-05-24 (a,e), the clear-sky measurements are indicated with open circles, whereas measurements in cloud fields are indicated with filled squares. Pink triangles represent the wind speed within updrafts (upper panels) and updraft contribution to the total flux (lower panels).

2 km) and thus were thicker than the clouds that were encountered on the eastern track. The thicker clouds do not only have larger momentum transport in the cloud layer, but also a much larger (percentage) contribution of the updraft to the total flux than any of the other measurements (Figure 6). The fraction of the leg that was occupied by updrafts was also significantly

larger than in all other cases. The deeper clouds may have been accompanied by wider updrafts with better protected cores that may be responsible for carrying larger fluxes.

Figure 10 and 11 show a time series of turbulence measured at 600 m in the mixed layer and at 1500 m in the cloud layer during the western track on June 4th. In grey the unfiltered turbulence statistics are shown, while black shows the linearly detrended and high-pass filtered ($f > 0.01$ Hz) statistics. Cloudy updrafts can have vertical speeds up to 5 m s$^{-1}$, in both

altitudes. Evidently, large buoyancy fluxes ($w'\theta'_v$, bottom row) are associated with large momentum fluxes, which reveals the importance of convection in generating a large momentum flux. Typically, updrafts carry wind speeds that are much slower (up to 5 m s$^{-1}$ for cloudy updrafts) than the environment.



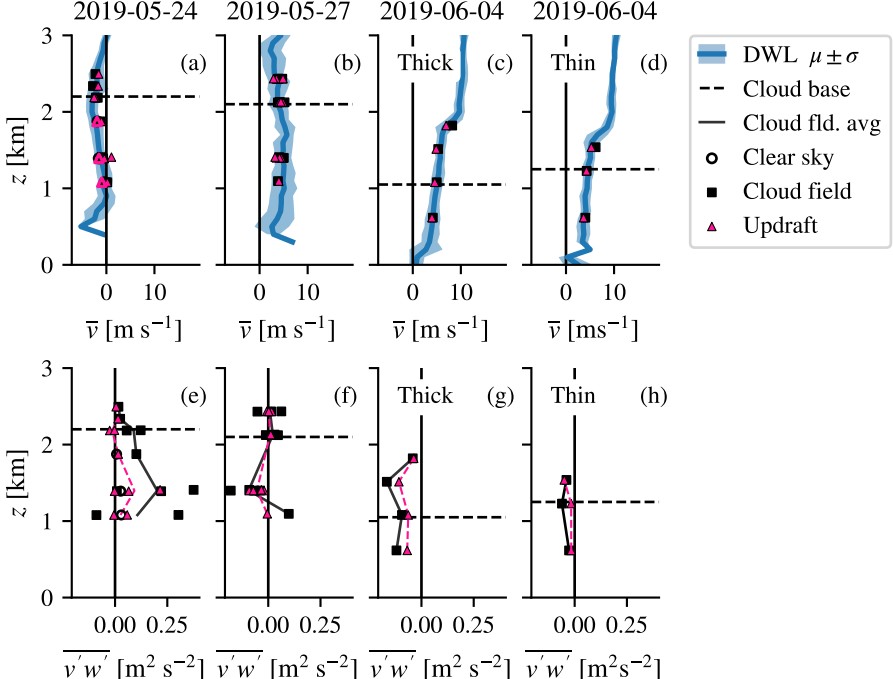

**Figure 8.** Same as Figure 7, but then for $v$ and $v'w'$. The v-axis is positive northward.

Looking carefully, one might see that in the mixed-layer $u'$ and $w'$ peak at different times and that $u'$ has a different sign in various updrafts (Figure 10). This could explain a much lower momentum flux. We discuss this further in the next sections, where we explore the fluxes sampled on (cloudy) updrafts, as well as how eddies of different scales contribute to the fluxes.

## 4.2 Scale contributions to flux

Large Eddy Simulations of various cases indicate that the momentum flux carried by small-scale shear-driven turbulent eddies (with a size smaller than $\sim 200$ m) can contribute more than 50% of momentum fluxes. Small scale turbulence may also transport momentum in an opposite direction than larger more coherent eddy structures (Zhu, 2015). This is particularly true for the lower mixed-layer and near cloud tops. However, in shallow cumulus cases, especially from the middle of the mixed-layer (sub-cloud layer) to the middle of the cloud layer, the net momentum fluxes are almost entirely carried by eddies with scales greater than 400 m.

In Figure 12, the total (net) momentum flux is shown for the legs in the sub-cloud layer, near cloud base, within the cloud layer and near cloud top for June 4$^{th}$. The momentum flux at different scales is calculated using a high-pass filter filters that with increasing cut-off frequency removes larger scales. The flux all the way to the right is for instance carried by eddies up to a scale of $\sim 7km$, corresponding to a cut-off frequency of 0.01 Hz. When the flux magnitude increases rapidly, it implies that



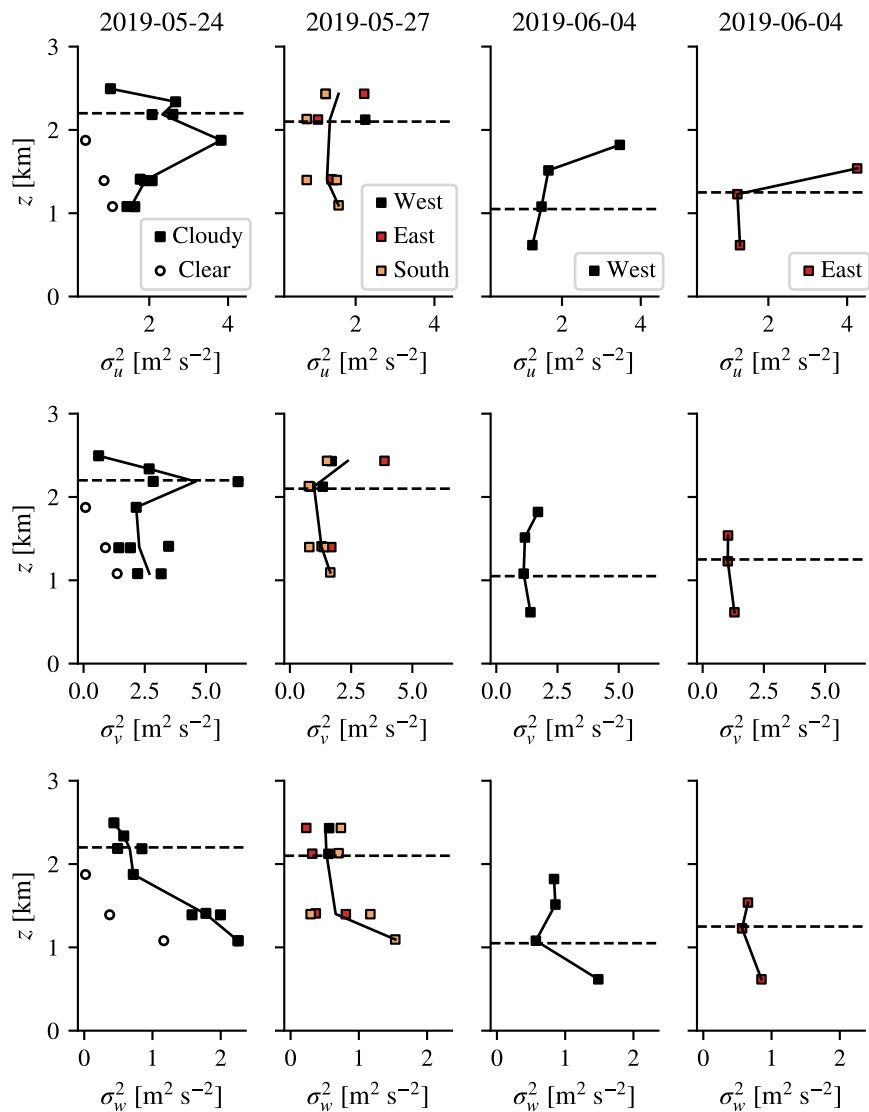

**Figure 9.** Variance of $u', v', w'$ for each flight.

larger-scale eddies contribute more to the momentum flux than smaller scales. If considerable flux is already at smaller scales (higher cut-off frequencies), as is the case at cloud top in the eastern leg, the smaller eddies play a more important role.

The results suggest that momentum fluxes carried by eddies of different scales can counteract to reduce the overall flux. The

relatively small flux (in the profiles, Figures 7,8) for instance in $\overline{u'w'}$ in the mixed layer on the western leg with thicker clouds, is produced by a positive $\overline{u'w'}$ carried by scales larger than 2.8 km up to maximally 7 km ($f_c = 0.025 - 0.01 Hz$), which almost compensates for the negative $\overline{u'w'}$ flux carried by turbulence on scales less than 2.8 km ($f_c = 0.025 Hz$).





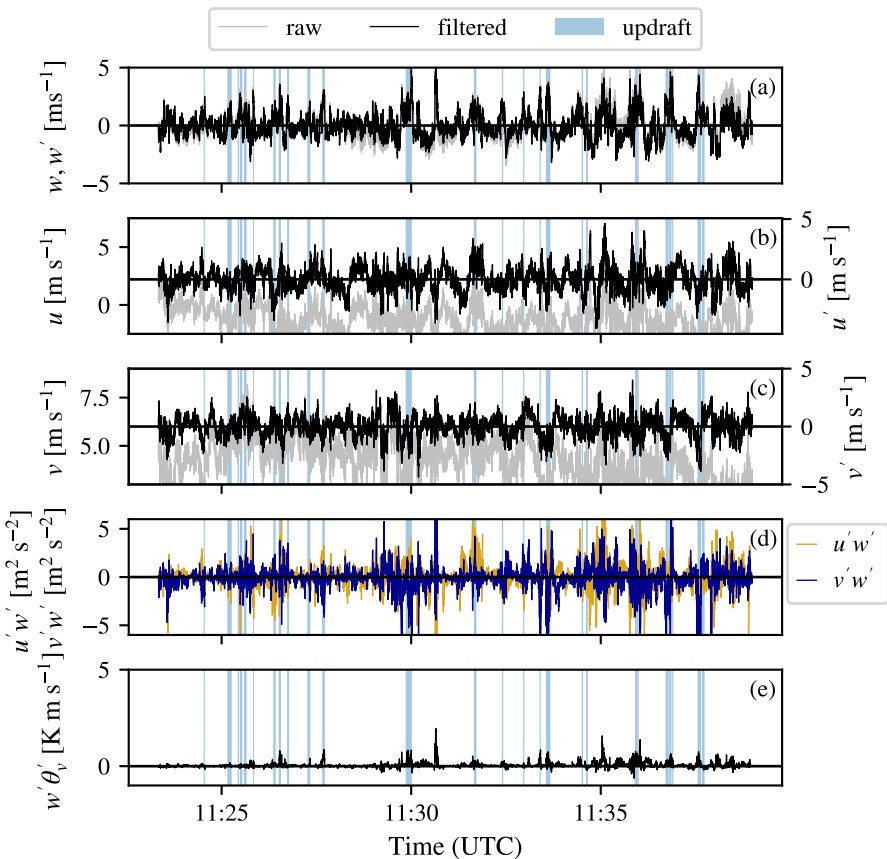

**Figure 10.** Raw (unfiltered) and fluctuating (filtered: linear detrended, high-pass filter cut-off 0.01 Hz) time series of (a) vertical velocity, (b) zonal wind, (c) meridional wind, (d) momentum fluxes, and (e) buoyancy flux, measured on 2019-06-04 at 600 m, in the middle of the mixed layer of the western leg. Updrafts are indicated with light-blue shading.

The same is true for $\overline{u'w'}$ near cloud top in the eastern leg with thinner clouds, and to a lesser extent, in the flux of $\overline{v'w'}$ in that leg within the mixed layer and near cloud tops. In the leg with thick clouds, the change in sign of the $\overline{v'w'}$ flux takes place

already between 0.7 - 2.8 km. In other words: the profiles deviate from a profile where fluxes linearly decrease with height when scales beyond 1-2 km play an important role.

## 5 Conclusions

In this paper we aimed to investigate the role of convection and clouds in setting the profile of momentum flux, guided by the questions: 1) How well can DWL observe the wind profile and wind variability? 2) How does the profile of the momentum flux

in different convective situations look like and is it in line with our theoretical expectation? And 3) Which scales contribute most to the momentum flux?

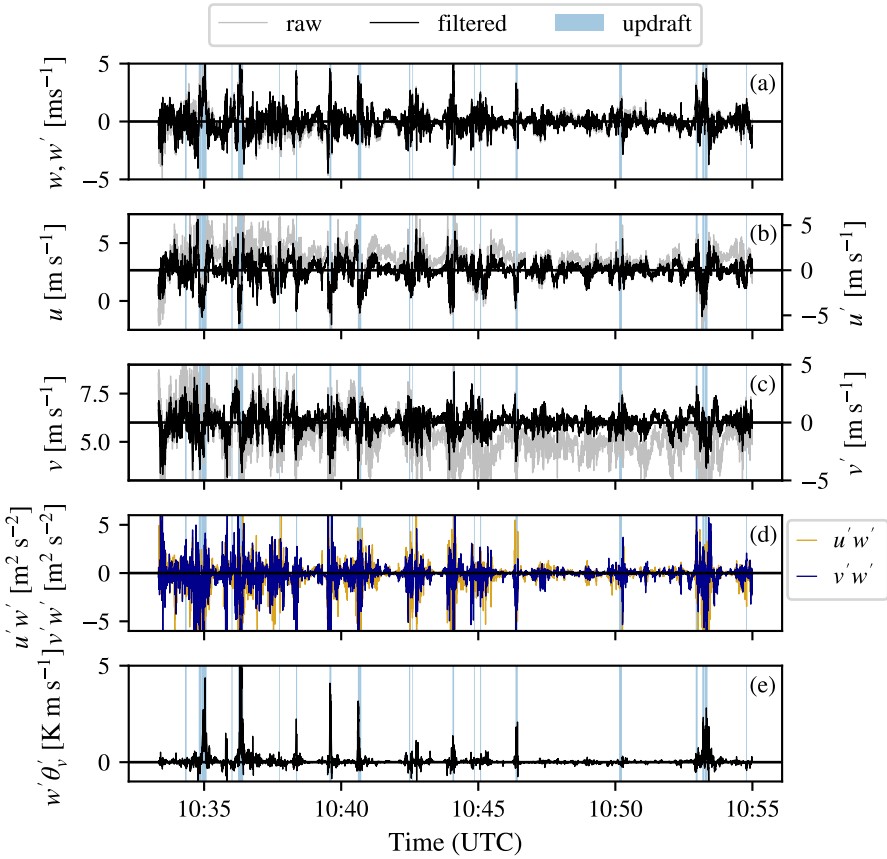

**Figure 11.** As in Figure 10 but for the leg at 1500 m, in the middle of the cloud layer. Updrafts are indicated with light-blue shading.

We address these questions using three case studies. The first case considers clear-sky and convective cumulus humilis conditions that developed over the Swabian Alps after a number of overcast and rainy days. Clouds were approximately 500 m thick and formed near an altitude of 2 km. Winds were quite calm, although a strong turning was present near 1.4 km and a temperature/moisture inversion was clearly present near 2.2 km. A second day provided less shear, both in speed and direction, but with much stronger winds. Thermodynamic structure and cloud base height were similar to the first flight, although many more clouds were present – also at mid- and higher altitudes. The approaching cold front needed us to move to keep targeting cumulus clouds. The last flight received most attention in our paper. There, clouds were randomly distributed and having many diverse cloud tops, strong increasing wind speed in the cloud layer, but much steady turning throughout the mixed-layer up to cloud top. Two tracks were flown with very similar thermodynamics as well as wind profiles, but with thicker clouds and lower cloud base on one of the tracks – an ideal situation to compare somewhat different clouds in similar conditions.

Below we will summarize our findings for the questions that we posed at the start of the study:



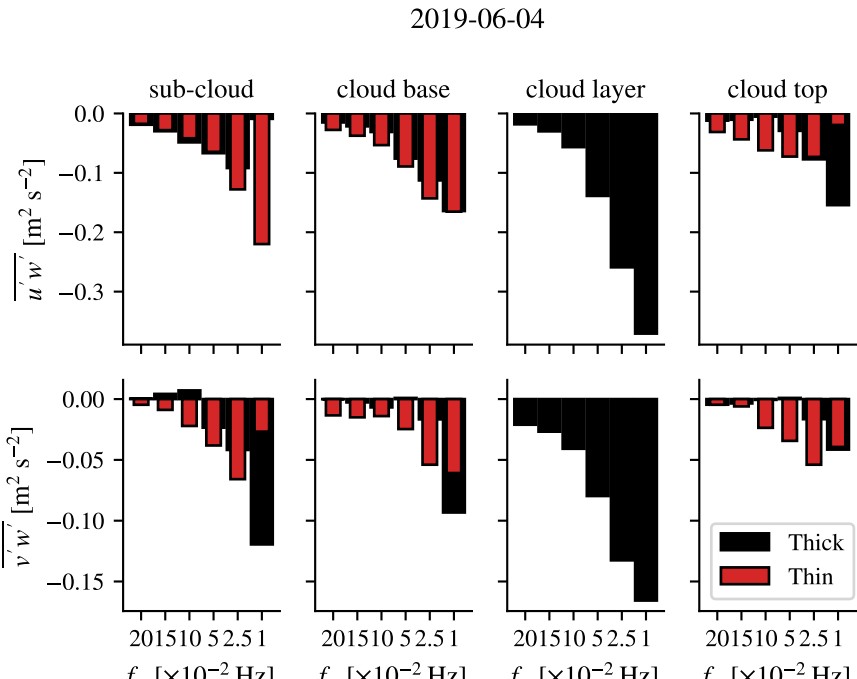

**Figure 12.** Total momentum flux $\overline{u'w'}$ (upper panels) and $\overline{v'w'}$ (lower panels) on the left, followed by the flux derived using a high-pass filter, with increasingly smaller filter scale (effectively including larger-scale turbulence). Filter scales are 10 Hz (keeping all scales $l <\sim 7$ m, assuming a cruising speed of 70 m s$^{-1}$), 1 Hz ($l >\sim 70$ m), 0.1 Hz ($l >\sim 700$ m), 0.01 Hz ($l >\sim 7$ km).

1. Comparing the Doppler Wind Lidar measurements to the in-situ "truth", we find that the leg means correspond well. Having a much larger resolution, 7 km opposed to 70 m, and much faster traveling speed, it is hard to quantitatively compare the DWL with the in-situ measurements. We find that the DWL is able to capture the mean and standard deviation of the horizontal wind components quite well, despite its much coarser footprint. This shows that the total variance in wind across a $\sim 100$ km transect is dominated by scales of several kilometres and larger.

   Furthermore, we benefit from the DWL profiles, as this allows us to better interpret the momentum fluxes using eddy diffusion and looking at the DWL anomaly values (measurement with subtracted mean), it shows the structure of the wind in the cloud layer and large parts of the mixed layer, as well as the location of up and downdrafts which revealed that negative vertical velocities that mark the top of the boundary layer as thermal plumes encounter a warmer environment and experience a negative buoyancy. They also show the presence of larger ($> 8$ km scale) structures in horizontal velocities with variations of the order of 3 m s$^{-1}$.

2. Most of the momentum flux profiles revealed down-gradient momentum transport that was generally strongest within the mixed-layer and decreasing towards cloud tops. On the same transect, flying from clear-skies to below cloudy skies



revealed a substantial enhancement in the total momentum flux measured (in the mixed layer $\sim$ -0.7 m$^2$s$^{-2}$ below clear sky compared to -0.3 m$^2$s$^{-2}$ below a cloud field).

During the third flight with post-cold-front and strongly sheared conditions with 1 km deep clouds the momentum flux profile did not decrease linearly with height as expected from shear-driven small-scale turbulence. The cloud field exhibited distinctive alternating patches of clear-sky and sheared clouds. Under this deeper convection we found that the total momentum flux was very small in the mixed-layer leg and large in the cloud layer. Here, the updraft contribution to the total flux has also been found much larger (percentual) than in any other case.

Horizontal momentum perturbations carried by updrafts within clouds measured up to almost 5 m s$^{-1}$ in the cloud layer on this flight. Under the deeper convection part, the fraction of updrafts that we encountered on the flight leg significantly increased, and they explained much larger part of the flux than in the other cases. These type of updrafts related to deeper clouds may already influence the wind speed in lower levels, and are important to consider in wind energy predictions.

3. Separating the different scales of the turbulence that contribute to the momentum flux show that small eddies can carry fluxes of different signs than larger thermally driven plumes or cells, overall leading to little total flux, or enhancing fluxes in the cloud layer where small-scale turbulence is small, thereby explaining deviations from a linearly decreasing flux profile.

*Data availability.* Data is made publically available on the 4TU Data Repository. DOI 10.4121/18614102 (currently reserved, will become public when item is published.)

*Author contributions.* M.K. and L.N. conceptualized the study, M.K. was responsible for the data analysis, and writing of the manuscript. C.M. was responsible for the technical preperation of the flight data and his experience aided the decision for an approach for eddy covariance estimation significantly. L.N. was also involved in the supervision. All authors provided critical comments on the quality of the work.

*Competing interests.* No competing interests are present.

*Acknowledgements.* This project has received funding from the European Research Council (ERC) under the European Union's Horizon 2020 research and innovation program (Starting Grant Agreement 714918).



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
