# Peer review of "Momentum fluxes from airborne wind measurements in three cumulus cases over land"

_Atmospheric Chemistry and Physics, 2022_

## Referee Comment (RC1)

General Comments about, "Momentum fluxes from airborne wind measurements in three cumulus cases over land," by Koning, Nuijens, and Mallaun

Reviewed by Margaret A. LeMone, Senior Scientist Emerita, NCAR

General Comments

The combination of lidar and aircraft measurements to estimate wind and momentum flux is wonderful to see – the lidar really puts the momentum fluxes in context.  For the Pennell/LeMone papers referenced in this paper, we had wind – but it was sampled by the aircraft taking the turbulence measurements, and there was some question as to whether the countergradient flux observed in the roll case was real, since the wind profile could have been evolving during the flight.

Something nearly equivalent was accomplished during ATOMIC/EUREC[4]A field campaign was with dropsondes to complement aircraft flux data – something once of the co-authors (Nuijens) knows about.

But the nearly-simultaneous measurements as done here is more flexible – one need only have the two aircraft and the necessary clearances.

The one big missing item in the paper is the nature of the terrain and land-surface properties beneath the flight tracks, which *can* have strong effects on fluxes and variances. More detail is needed on the surface and the flight patterns themselves.  (The tracks are helpful – but "deviations" from tracks should be described in more detail – some comments about this are in the specific comments).

I think it would be helpful to the reader to at last know the location, time, and goals of the CloudBrake campaign in the introduction.  One sentence is enough.

While Fig. 1 is extremely helpful, It would be also be helpful to have a table that briefly summarize each flight, so that the unfamiliar reader can quickly flip the pages to find it, rather than hunting through the text to understand each case.  This could also be the place to define "thick" and "thin" – which should be consistently used in the figures (Sometimes "east" and "west" are used instead.)

Finally, in relating momentum fluxes beneath cloudy areas to those in clear areas, it might be of interest to examine the buoyancy fluxes.  This is not a requirement – but may, along with terrain and land cover, be a factor in determining horizontal distribution of fluxes.

Specific Comments

1.  L31.  "Convection and clouds" … since clouds are convection, not sure what this means.  Are you referring to cloud- and subcloud-layer convection?  Or "dry and moist" convection?

2. L42.  I think you mean here Pennell and LeMone, which has profiles of momentum fluxes through the cloud layer, unless you are referring to LeMone and Pennell's Figure 5.

3. L45.  Rather than small-scale – are you referring to dry-air convection?   In LeMone and Pennell, Fig. 5, rolls accounted for a significant amount of the momentum transport.  In this case, the clouds were extremely shallow.  The linear flux dependence disappeared when clouds became significant (Case III).

4. L50, 52. Please define 'mesoscale.'  km scale?

5. L77.  Year?  Then it doesn't need to be repeated.

6. L88. Flights adjusted to capture cumulus clouds.  What does this mean … deviations from a straight line?  (A zig-zag pattern?  Movement of the entire track?)

7. Fig 1 and discussion.
   a) Were the flight tracks the same for the Falcon and the Cessna? (Refer to Fig. 1 on this.)
   b) I am assuming Cessna and Falcon flight legs were designed to overlap in time and space to the degree possible.  Is this correct?
   c) Please describe surface conditions (terrain, significant vegetation variation) beneath the tracks
   d) Please include the typical data-collection speed of the two aircraft here, even though they are given later.
   e) Finally, you might mention that the flight legs were flown crosswind, which can have impact on the sample, particularly in stronger winds.

8. L91 (Just below Fig. 1). … "near cloud top" … above cloud top?   Or just below the top of the highest clouds?

9. L102.  Cruising speed?  Not necessary if mentioned earlier – this is the first question I had when I saw the frequency range.

10. L129.  Along-beam? (Rather than vertical?).  Vertical resolution is along-beam resolution x cos (20 degrees).

11. L130.  Pulse length?

12. L134.  Presumably the 8 km applies to a specific vertical distance.  Nearer the aircraft, the resolution would be better.

13. L145, bottom of p. 6.  "turning points between legs."  180 degrees?  90 degrees? Fig 1 shows two parallel flight tracks for two of the days, and three tracks for the third. Were four levels flown by the Cessna along both flight tracks?  Did the Falcon do reverse-heading legs

on the west leg and then fly east leg for reverse headings?  Or did it do U or box patterns? Minor stuff – but it can affect interpretation.

14. Bottom of p. 6.  "Horizontal resolution 8 km" … Is this because the width of the cone at Cessna flight level roughly 8 km?   How many VAD circles are executed for each 8 km the Falcon travels?  For each flight leg?  I am assuming you are just getting wind vectors.

15. L154 and L155.  "slightly overestimated."  Not sure what this means.  The variances are over different scales, aren't they? (Unless you are estimating variances using VAD as well, which doesn't seem to be the case from L133-4, and if so – aren't the scales represented still larger than what the aircraft sees?).  Wouldn't it be more precise to say simply that the variance of the 8-km averaged wind is greater than the variance of the wind measured by the aircraft?

16. L157.  1-2 km is the "effective" horizontal resolution of the DWL?  I thought it was 8 km! Please explain in section on lidar.

17. L160.  Just out of curiosity, what sort of magnitudes do you get for mean vertical velocity from the lidar?  That is such a difficult measurement – and of course you would have to know the aircraft vertical velocity as well.

18. L165.  How well does the Lenschow-Stephens method work in cloud?  In studies over the tropical oceans, we found that using vertical velocity alone worked better, partially because measurements of temperature and mixing ratio in cloud were not that accurate (Series of papers by LeMone and Zipser and others).  Might consider trying this in the future. (Continental clouds may work better than tropical oceanic clouds – and mixing-ratio instruments could be more reliable).

19. L172-4.  This makes sense.  We found stronger subcloud and lower cloud-layer vertical velocity standard deviation in more cloudy conditions (Fig 11, Nicholls and LeMone, JAS, 1980), which makes sense, since associated buoyancy field and/or interaction with the shear generates pressure perturbations that can draw up air from below (LeMone et al. Mon. Wea. Rev. Feb and Oct 1988), also see Rotunno and Klemp 1982).  This might be something to look at in a future paper.

20. Fig. 5.  If U and V were plotted rather than direction and speed, it would be easier to relate them to the fluxes, and the wind turning on 24 May would show up in only slight changes in the wind components.  (I see that they are plotted later in Figs. 6 and 7, and that the resolved winds do show strong shear – is it real?)  So maybe no modification needed here.

21. L231. Skewness is not shown in Fig. 9. Delete "and skewness of"?

22. L249. "Cloudy updrafts can have vertical speeds … in both altitudes." You mean the updrafts in the subcloud layer are clearly associated with individual clouds?  Or you mean updrafts

beneath cloudier areas?  (Two reasons for this comment – the possible impact of terrain  – though I recognize it might be unlikely, and that 600 m is in the subcloud layer).

23. L252.  Same comment

24. L257, Scale contribution to flux.  Near the surface, Kaimal, Wyngaard, Izumi, and Coté (1972, QJRMS) found that the cospectra of $\overline{u'w'}$ and other fluxes follow a fixed slope, with large scales more significant. Of course, things could differ from this significantly higher up, as possible in cases with quasi-two dimensional convection like clear-air roll vortices, and clouds, as is noted in this paper.

25. L268.  Fig. 9 should be labeled like the others – thick and thin.  Although there seems to be a "south" here as well.  Are the fluxes significantly different on the "south" leg?

26. L276.  Greater than 1-2 km?

27. L286.  It looks like mixing ratios were higher on the cloudier day.

28. L287.  Again – should specify what is meant by,"The approach cold front needed us to move to keep targeting cumulus clouds."  Did the whole track get moved, or was more of a zig-zag pattern flown?

29. Figure 12. caption, line 3.  "Effectively **Ex**cluding"? In the figure, you have six bars suggesting six filter scales, but in the caption, there are only four filter scales.  Shouldn't these be consistent?

30. L295.  Aren't the standard deviations of horizontal wind from the aircraft for much larger scales than the aircraft, which measures the standard deviation on turbulence scales?  (i.e., 8 km (or a few km?) and larger vs 7 km and smaller).   Could the nearby mountains contribute to this larger-scale variance in wind, e.g., though mountain related periodic waves? (as well as the distance between clouds)

31. 305.  Enhancement of fluxes below clear skies?  On L172-174 the opposite was stated.  Or is this a different day? (Such a situation is possible, either due to enhanced heating of the ground or to the impacts of terrain or ground cover).

32. L311… "very small" … $\overline{u'w'}$  was nearly zero, but there was significant $\overline{v'w'}$ … perhaps just 'smaller'?

---

## Author Comment (AC1)

**General response letter:**

Dear Margaret and Chris, dear Editor,

We were very pleased to read your honest and open reviews of our manuscript. Thank you very much for taking the time to prepare your review. We also appreciated your acknowledgement of the value of observations of wind and momentum fluxes, which are not always at the forefront of studies on convection.

You raised two main points of concern. The first is that we should include more information on the underlying land surface that can play an important role in setting surface momentum exchange and momentum fluxes higher up. There is no real excuse for not paying more attention to the surface, a neglect probably caused by spending much time thinking about convection over oceans. We included more information about the terrain, as outlined in more detail below.

Second, you mentioned that the manuscript describes the different situations but lacks synthesis or a discussion on what generic findings are applicable to all flights. It would be fair to say that we struggled with this. The limited number of flights allows at most to demonstrate a concept and raise questions for further investigation. Yet there are a few important points we had thought of demonstrating and in revising the manuscript we attempt to communicate the following:

- There is significant (meso-gamma (2 – 20 km) scale variability in the wind with changes over 5 m/s that are captured by the downward looking wind lidar.
- Momentum fluxes increase in areas with (cloudy) updrafts, but the contribution of the updraft to the total momentum flux typically a third to two-thirds, which is much less than the contribution of the updraft to buoyancy flux.
- Scales beyond 1 km contribute significantly to the momentum flux and there is clear evidence for compensating flux contributions from different scales.
- Different flight segments, even on the same flight day, can have a very different momentum flux profile that may not be explained by turbulent transport across local vertical gradients in wind.

All of this highlights that momentum flux profiles and their variability require understanding of motions across a range of scales, with non-negligible contributions of the clear-sky fluxes and of mesoscales that may be coupled to the convection. Wind lidars can help elucidate the flows on larger than cloud scales and should be used more deliberately in studies of clouds and their spatial organization.

To better bring out the above points, we implemented the following changes:
- Revised figures with additional information, such as the total and updraft-carried buoyancy flux profile (in a new figure), profiles of wind variance as seen by the wind lidar (in Figure 9 of the manuscript, Figure 10 in the revised manuscript), the dependence of cross-variance (momentum flux) ánd variance on filter scales on a day with no shear (May 27) in addition to a day with strong shear (June 4) (in Figure 12, Figure 11 in the revised manuscript).
- We omitted one of the time series displaying turbulence and added two panels that show the diverging and converging motions in the east-west wind component, demonstrating that circulations are present that are tied to convection.
- We revised the abstract, introduction and conclusions to better reflect these points, along with changes in wording throughout the text.

We believe these changes are valuable additions and hope you accept them. Below you can find specific answers to individual points you raised.

Sincerely,

Mariska Koning, Louise Nuijens and Christian Mallaun

Point by point reply to Margaret LeMone

On the terrain description
* * *
We added the terrain description in the text.

The terrain below was mostly used for agriculture with low crops, occasionally encountering patches of trees or villages. On the first two flights a hilly topography was present, whereas the last flight was above flat land.

Specific Comments
* * *
1. L31. "Convection and clouds" … since clouds are convection, not sure what this means. Are you referring to cloud- and subcloud-layer convection? Or "dry and moist" convection?

We referred to dry and moist convection, which we made explicit in the text now.

Not only dry convection, but also moist convection plays an important role in this process, because clouds extend the boundary layer height, tapping in regions aloft with faster moving winds. This transport of momentum (momentum fluxes) by convective eddies (thermals) and through clouds is broadly called convective momentum transport (CMT).

2. L42. I think you mean here Pennell and LeMone, which has profiles of momentum fluxes through the cloud layer, unless you are referring to LeMone and Pennell's Figure 5.

We do refer to LeMone and Pennell's Figure 5 (which is indeed the momentum flux profile).

3. L45. Rather than small-scale – are you referring to dry-air convection? In LeMone and Pennell, Fig. 5, rolls accounted for a significant amount of the momentum transport. In this case, the clouds were extremely shallow. The linear flux dependence disappeared when clouds became significant (Case III).

This paragraph has been changed entirely, taking into account your comments:

"Our understanding of turbulent wind fluctuations throughout the boundary layer largely stem from a handful of in situ turbulence measurements during research aircraft fights at selected height levels in subtropical settings. A seminal study is that by LeMone and Pennell (1976), where flight tracks below and through cumulus fields near Puerto Rico were used to derive wind and flux profiles. This work highlighted that the momentum flux profile can take a very different shape depending on clouds overhead. In particular, they found that in fields of cumulus clouds organized in rolls, the rolls were responsible for a significant amount of the momentum transport even though clouds were extremely shallow. In fields of more significant and randomly distributed clouds, the linear flux dependence disappeared, becoming counter-gradient at various altitudes. However, some doubt

remained as to whether the wind profile could have evolved during the flight, because the profile itself was only sampled by the turbulence measurements at selected legs."

4. L50, 52. Please define 'mesoscale.' km scale?

Three prototype flights were carried out focusing on measuring the wind environment in convective situations to evaluate turbulent to mesoscale (up to 7 km) wind fluctuations and implications for the momentum flux profile.

5. L77. Year? Then it doesn't need to be repeated.

I've added the measurement year of 2019.

6. L88. Flights adjusted to capture cumulus clouds. What does this mean … deviations from a straight line? (A zig-zag pattern? Movement of the entire track?)

This means that we changed the originally planned tracks and started legs at a different location. All flown locations are shown in Figure 1. The text is adapted to clarify this statement.

7. Fig 1 and discussion.
a) Were the flight tracks the same for the Falcon and the Cessna? (Refer to Fig. 1 on this.)

Yes they were. We added "the same" in the sentence: "During the 2--2.5-hour-flights, the two aeroplanes flew back and forth across **the same** pre-defined tracks."

b) I am assuming Cessna and Falcon flight legs were designed to overlap in time and space to the degree possible. Is this correct?

Yes they were. Adapted in text, by adding sentence: "During the 2--2.5-hour-flights, the two aeroplanes flew back and forth across the same pre-defined tracks. Because the two planes have different cruising speed (the Cessna about 70 m s$^{-1}$, for the Falcon about 200 m s$^{-1}$), the pre-defined tracks ensure overlap in space and time to the degree possible."

c) Please describe surface conditions (terrain, significant vegetation variation) beneath the tracks

We added the following description in the text: "The terrain below was mostly used for agriculture with low crops, occasionally encountering patches of trees or villages. On the first two flights a hilly topography was present, whereas the last flight was above flat land."

d) Please include the typical data-collection speed of the two aircraft here, even though they are given later.

Cessna 100 Hz, Falcon 1/40 Hz. The text has been adapted accordingly (marked fat below):

"Turbulence measurements using an in situ (3D) **turbulence probe aboard the DLR Cessna Grand Caravan were taken at 100 Hz** along that track at four different altitudes: within the mixed-layer, near cloud base, within the cloud layer and through the tops of only the thickest clouds. Employing the downward staring **Doppler wind LiDARs at a measurement rate of 40 s**, the DLR Falcon remained around 11 km altitude throughout the flight. The instruments are described next."

e) Finally, you might mention that the flight legs were flown crosswind, which can have impact on the sample, particularly in stronger winds.

We agree to your comment. We have mentioned that we flew mostly crosswind in the manuscript.

8. L91 (Just below Fig. 1). … "near cloud top" … above cloud top? Or just below the top of the highest clouds?

Near cloud top is indeed unclear. We flew through cloud tops of some of the thickest cumuli, meaning that we missed most of the cloud tops as many clouds did not extend this high.

Changed sentence to : "within the mixed-layer, near cloud base, within the cloud layer and through the tops of only the thickest clouds"

9. L102. Cruising speed? Not necessary if mentioned earlier – this is the first question I had when I saw the frequency range.

We added it to the text. See text changes in answer 7b

10. L129. Along-beam? (Rather than vertical?). Vertical resolution is along-beam resolution x cos (20 degrees).

Added: "(i.e. along-beam resolution approx. 94 m)".

11. L130. Pulse length?

Instead of pulse width, I presume. Liu talked about pulse width, which is why I called it that way. Length does make more sense, so I changed it. Thank you!

12. L134. Presumably the 8 km applies to a specific vertical distance. Nearer the aircraft, the resolution would be better.

The 8 km is purely based on the moved (ground-)distance between two full-scans. Because it takes about 40 seconds to complete a scan, and the aircraft moves at a speed of 200 m/s, a distance of 8 km is completed before the new scan is started. Averaging each line-of-sight measurement, one measurement is thus representative of those 8 km. The volume/area of the measured air mass must indeed be smaller nearer the aircraft than below, due to the cone measurement (having a cone angle of 30 degrees).

We changed to "… the horizontal resolution (distance traveled between two measurements) is about 8 km."

13. L145, bottom of p. 6. "turning points between legs." 180 degrees? 90 degrees? Fig 1 shows two parallel flight tracks for two of the days, and three tracks for the third. Were four levels flown by the Cessna along both flight tracks? Did the Falcon do reverse-heading legs on the west leg and then fly east leg for reverse headings? Or did it do U or box patterns? Minor stuff – but it can affect interpretation.

a) *turning points* Thanks for this question. There is indeed a transfer from track at 11:46 UCT. The second event is a return point at the same leg. The picture is now adapted to show a longer time series. We now also made sure not to include the transfer from the western to the eastern leg. With turning points I mean the plane turns 180 degrees, doing reverse headings. The text is adapted to clarify this.

b) The Cessna did not fly at all four levels at all flown tracks (although we did design the flight that way before executing them). This is visible for 4 June 2019 in the eastern leg. On May 29th, the west track only flew at deepest cloud tops and cloud layer, the east track flew in the cloud layer and at twice in the mixed layer, whereas the south track measured at two heights in the mixed layer, at cloud base and at cloud top.

c) Falcon flew reverse-headings legs on the same tracks at the same time as the Cessna.

14. Bottom of p. 6. "Horizontal resolution 8 km" … Is this because the width of the cone at Cessna flight level roughly 8 km? How many VAD circles are executed for each 8 km the Falcon travels? For each flight leg? I am assuming you are just getting wind vectors.

Indeed, we just got the wind vectors. There is one VAD circle completed for each 8 km that the Falcon travels. Terminology was taken from Witschas et al. (2020). Would it be better to call it measurement interval distance?

15. L154 and L155. "slightly overestimated." Not sure what this means. The variances are over different scales, aren't they? (Unless you are estimating variances using VAD as well, which doesn't seem to be the case from L133-4, and if so – aren't the scales represented still larger than what the aircraft sees?). Wouldn't it be more precise to say simply that the variance of the 8-km averaged wind is greater than the variance of the wind measured by the aircraft?

We followed the advice to say variance or mean in DWL is greater/smaller than that of the in-situ measurements. Indeed, we do estimate the variances from the provided u,v-components.

16. L157. 1-2 km is the "effective" horizontal resolution of the DWL? I thought it was 8 km! Please explain in section on lidar.

We meant to say that the wind fluctuations are dominated by scales that the DWL with a resolution of 8 km can capture, so that scales associated with cloud convection are not the main cause of horizontal wind fluctuations. We changed the text:

"This gives us confidence that the DWL can provide complementary information of the (horizontal) wind profile at heights where in-situ measurements are absent. It also tells us that horizontal wind fluctuations are largely set by scales of 8 km or larger and that cloud convection scales of 1-2 km is less important."

It also tells us that horizontal wind fluctuations are dominated by scales larger than 1-2 km (the effective horizontal resolution of the DWL is ∼ 8.4 km).

17. L160. Just out of curiosity, what sort of magnitudes do you get for mean vertical velocity from the lidar? That is such a difficult measurement – and of course you would have to know the aircraft vertical velocity as well.

Of course, the magnitude slightly changes with height. Therefore, I made this plot showing the quantiles of 5%, 25%, 75%, 95% for each height. At the bottom we see more spread, but this may be due to smaller sample size.

[Figure]

18. L165. How well does the Lenschow-Stephens method work in cloud? In studies over the tropical oceans, we found that using vertical velocity alone worked better, partially because measurements of temperature and mixing ratio in cloud were not that accurate (Series of papers by LeMone and Zipser and others). Might consider trying this in the future. (Continental clouds may work better than tropical oceanic clouds – and mixing-ratio instruments could be more reliable).

We do not have a cloud indicator data, which makes it difficult to be sure about this. To detect clouds we can look at locations of saturation, which is complicated because often the measurements indicate close to saturation and not 100% saturation. We did also have a cloud selecting algorithm, taking a 99% relative humidity as threshold for cloud presence and minimum cloud size of 100 m (to be consistent with the updraft selection).

Bearing this in mind, we see that in the cloud layer, our cloud algorithm picks up less events than the updraft selection and a few that are around the same location, but not exactly. However, for consistency and to be able to say something about the sub-cloud layer, we chose for the updraft selection. This may result in a lower contribution to the flux in the cloud layer than when using the cloud selection.

19. L172-4. This makes sense. We found stronger subcloud and lower cloud-layer vertical velocity standard deviation in more cloudy conditions (Fig 11, Nicholls and LeMone, JAS, 1980), which makes sense, since associated buoyancy field and/or interaction with the shear generates pressure perturbations that can draw up air from below (LeMone et al. Mon. Wea. Rev. Feb and Oct 1988), also see Rotunno and Klemp 1982). This might be something to look at in a future paper.

We added the reference of Nicholls and LeMone and will keep this mechanism in mind for future research.

20. Fig. 5. If U and V were plotted rather than direction and speed, it would be easier to relate them to the fluxes, and the wind turning on 24 May would show up in only slight changes in the wind components. (I see that they are plotted later in Figs. 6 and 7, and that the resolved winds do show strong shear – is it real?) So maybe no modification needed here.

21. L231. Skewness is not shown in Fig. 9. Delete "and skewness of"?
This has been removed accordingly.

22. L249. "Cloudy updrafts can have vertical speeds … in both altitudes." You mean the updrafts in the subcloud layer are clearly associated with individual clouds? Or you mean updrafts beneath cloudier areas? (Two reasons for this comment – the possible impact of terrain – though I recognize it might be unlikely, and that 600 m is in the subcloud layer).

We changed the cloudy updrafts to just "updrafts" so that should have been changed before.

Addressing the question: We wanted to say that the maximum vertical velocity at 600 m (in the middle of the mixed-layer) is similar at 1500 m (in the middle of the cloud layer) as is the slower horizontal wind speed that is measured at the same location as these updrafts. We cannot relate the mixed-layer measurements to the clouds overhead, as we did not have the possibility to measure/look at the clouds overhead during the measurements in the mixed-layer.

We removed one of the graphs following the advice of Chris Fairall (who also reviewed this article), and changed the text accordingly.

23. L252. Same comment

See previous answer.

24. L257, Scale contribution to flux. Near the surface, Kaimal, Wyngaard, Izumi, and Coté (1972, QJRMS) found that the cospectra of $u'w'$ and other fluxes follow a fixed slope, with large scales more significant. Of course, things could differ from this significantly higher up, as possible in cases with quasi-two dimensional convection like clear-air roll vortices, and clouds, as is noted in this paper.

This is indeed a nice addition to the discussion, and we included this statement and a reference to the paper by Kaimal et al. (1972).

25. L268. Fig. 9 should be labeled like the others – thick and thin. Although there seems to be a "south" here as well. Are the fluxes significantly different on the "south" leg?

On the second day 2019-05-27, we had three tracks: East, west and south. On 4 June, we had an east and west track that had different cloud thickness. We tried to clarify the labels throughout the paper.

26. L276. Greater than 1-2 km?

Rephrased: In other words: the profiles deviate from a profile that is dominated by diffusive turbulence, where fluxes would linearly decrease with height.

27. L286. It looks like mixing ratios were higher on the cloudier day.

Conclusions were changed – not applicable anymore. But indeed, mixing ratios were higher on the cloudier day.

28. L287. Again – should specify what is meant by,"The approach cold front needed us to move to keep targeting cumulus clouds." Did the whole track get moved, or was more of a zig-zag pattern flown?

Conclusions were changed – not applicable anymore.

29. Figure 12. caption, line 3. "Effectively Excluding"? In the figure, you have six bars suggesting six filter scales, but in the caption, there are only four filter scales. Shouldn't these be consistent?

a) *effectively excluding*

To improve this point, the caption is changed to: "Scale contributions to the momentum flux u'w' (upper panel) and u'v' (lower panel) for different heights in the atmosphere. The bars in each panel represent the flux contribution, in which the left-most bar only includes small scales (frequencies exceeding 0.2 Hz) and the right-most bar includes all scales that are represented in the measurements limited to 7 km (with a cut-off frequency of 0.01 Hz, excluding lower frequencies). Filter scales are 0.20, 0.15, 0.10, 0.05, 0.025, and 0.001 Hz that, when assuming a cruising speed of 70 m/s, correspond to length scales of 350, 467, 700, 1400, 2800, and 7000 m. "

b) *filter scales in caption* You are right. Thank you for pointing this out.

The text has been changed: "Filter scales are 0.20, 0.15, 0.10, 0.05, 0.025, and 0.001 Hz that, when assuming a cruising speed of 70 m $s^{-1}$, correspond to length scales of 350, 467, 700, 1400, 2800, and 7000 m."

30. L295. Aren't the standard deviations of horizontal wind from the aircraft for much larger scales than the aircraft, which measures the standard deviation on turbulence scales? (i.e., 8 km (or a few km?) and larger vs 7 km and smaller). Could the nearby mountains contribute to this larger-scale variance in wind, e.g., though mountain related periodic waves? (as well as the distance between clouds)

The wind in the boundary layer will always be influenced by the terrain in the surrounding. The synoptic conditions, different gravity waves and turbulence constitute the wind. Likely differential heating by the hilly area had an influence on the cloud formation and turbulence scales that constitute the winds in this study.

31. 305. Enhancement of fluxes below clear skies? On L172-174 the opposite was stated. Or is this a different day? (Such a situation is possible, either due to enhanced heating of the ground or to the impacts of terrain or ground cover).

This is not well-phrased. Meaning that transitioning from clear-skies towards cloudy skies, we see an increase in momentum flux. The numbers however, suggested otherwise. Checking them, a missing leading zero was found: 0.07 $m^2s^{-2}$ below clear skies, but -0.3 $m^2s^{-2}$ below cloudy skies.

Conclusions were changed – not applicable anymore.

32. L311… "very small" … $u'w'$ was nearly zero, but there was significant $v'w'$ … perhaps just 'smaller'?

Conclusions were changed – not applicable anymore.

Point by point reply to Chris Fairall

*Figure 2 would be more illuminating to turbulence people is the streamwise, cross stream, and vertical spectral components were presented on the same graph. Perhaps a 4-panel figure with a panel for a different height. Also, suggest plotting frequency*Spectrum to be area conserving in log-log space.

Thank you for your remark. We adjusted the figure. Please do note that the u, v components are almost along the streamwise and cross stream direction, so changes are not very large.

*Computation of fluxes via eq 1 is equivalent sampling the time series with a square window, computed the cospectra, and integrating that to get <w'x'>. The authors use Hann window for their variance spectra, which has advantages over the square window. I suggest they use Hann or Hamming window for flux computation. This will reduce leakage from lower frequencies.

Before computing the eddy covariance flux, we applied a high-pass filter to the data with a cut-off frequency of 0.01 Hz in order to remove effects from larger scales. We were not stating this in the dedicated paragraph, so we did add this there. This approach for calculating fluxes was taken, so that we could also take out the updraft locations to calculate the average flux in these areas.

*Eq 2 introduces the K coefficient but the authors don't do anything with it except to say fluxes tend to be down gradient. I would not mind seeing some values and relationship to sigma_w and/or the scale associated with the peak of the vertical velocity spectrum.

Thank you for your comment. We decided to remove the equation as it is not of much use and may cause different expectations. During our analysis we have calculated the values of K backwards, which results in a wide range of magnitudes. Depending on the amount of shear and flux magnitude, we get very high numbers when there is some flux but almost no shear and very low K-values when fluxes are small and shear is larger. This is because K-diffusion is a way to estimate the flux from the wind gradient profile. Therefore, this may not be the intended use for this equation. We replaced it with a small text to illustrate the idea, but not to focus too much attention to the equation/theorem:

"When fluxes are dominated by small-scale turbulent diffusion, it may be modeled (parametrized) by using so-called flux-gradient relationships of K-diffusion. We find that most of the fluxes and their relationship with the wind gradient lead to a K-value that is in line with down-gradient diffusion, acting to reduce the wind gradient."

The other point you raised was about the peak in the w-spectrum that likely resides at frequencies that are lower than we can see. We do not have enough statistics to look at even lower scales. We are in the energy cascade frequencies, being unable to capture the energy generating eddies that probably have even lower frequencies (larger scales), because we do not have enough statistics.

*I did not find Figs. 10 and 11 to be that helpful. Perhaps they could be dropped.

We do believe it insightful to have a view on the time series and the outcome of the updraft selection. However, two figures may be too much. We removed the figure showing time series of a leg in the cloud layer.